# WEBARENA VERIFIED

## ABSTRACT

Autonomous web agents have been growing in interest within the research community and are increasingly operating in more complex multistep browser workflows. However, widely used benchmarks can misestimate performance due to underspecified goals and fragile evaluators—challenges typical of normal benchmark maturation rather than flaws in the paradigm. This hinders accurate performance assessment and limits benchmark utility for guiding method development. We present WebArena Verified, a reproducible re-evaluation of WebArena that retains its containerized environments while strengthening measurement. We audit all 812 tasks, repair misaligned checks and clarify ambiguous instructions, replace substring matching with type-aware exact matching with semantic normalization, verify backend state for mutation tasks, and adopt a structured JSON schema with explicit status codes for deterministic scoring. For reporting, we provide per-template macro-averaged metrics, 95% confidence intervals, and failure-mode breakdowns. Our new evaluator reduces the false-negative rate by 11.3 percentage points compared to the original scoring pipeline. To reduce evaluation overhead, we introduce WebArena Verified Hard, a 258-task subset that retains difficult tasks, reduces runtime by 68.2%, and maintains discriminative power and coverage. WebArena Verified remains drop-in compatible with existing agents, requiring only minimal changes and enabling faithful comparisons. Code, data, and evaluation tools will be released to support reproducibility.

## 1 INTRODUCTION

Autonomous web agents are now capable of executing increasingly complex web workflows that demand dynamic navigation and extraction of visual and textual information. As these systems are adopted more widely, robust and reproducible evaluation becomes essential for meaningful progress. Existing benchmarks provide a solid foundation for measuring these abilities. For instance, WebArena (Zhou et al., 2024) offers self-hosted, containerized sites that emulate real-world environments, permitting task execution through standard browsers. The design of WebArena inspired a family of follow-up benchmarks—VisualWebArena (Koh et al., 2024), Mind2Web (Deng et al., 2023), WorkArena and WorkArena++ (Drouin et al., 2024; Boisvert et al., 2025), OSWorld (Xie et al., 2024), and AndroidWorld (Rawles et al., 2024)—each expanding the scope of realistic tasks and interface modalities. However, widespread adoption has revealed systematic evaluation errors that undermine measurement validity (Zhu et al., 2025; Koh et al., 2024). Zhu et al. (2025) conducted a recent audit of 37 public agent-benchmark suites—covering 13 web-interaction domains—that revealed pervasive quality issues that erode confidence in reported results due to underspecified success criteria, allowing disparate interpretations of task completion or low precision evaluators. For example, on $\tau$-bench (Yao et al., 2024), a trivial "empty-output" agent achieved 38% success, outperforming GPT-4o-based agents on intentionally impossible tasks. On WebArena, the brittle string matching inflates success rates by 1.4–5.2% (Zhu et al., 2025). These issues reflect broader patterns in benchmark maturation observed across machine learning benchmarks, yet they must be addressed. Without robust evaluation, researchers struggle to identify genuine performance gaps, increasing the risk of over-optimistic conclusions, and benchmark-specific adaptations while substantive deficiencies remain unaddressed. These reliability concerns have sparked efforts to improve evaluation quality through principled debugging and systematic verification, such as SWE-Bench Verified (Chowdhury et al., 2024), OSWorld Verified (Xie et al., 2025) and WorkArena++ (Boisvert et al., 2025). This pattern of "verified" re-releases demonstrates the value of strengthening evaluation

mechanisms while preserving benchmark utility and adoption. For example, on SWE-bench Verified, agents roughly doubled performance once flawed tasks were removed (Chowdhury et al., 2024).

We introduce **WebArena Verified**, an updated version of WebArena that preserves the original containerized environments while improving evaluation reliability. Our approach re-verifies task definitions, reference answers, and evaluators using deterministic evaluation with explicit success criteria. We replace brittle string matching with backend state verification, standardize outputs to a structured JSON schema, and report template-level macro-averages with 95% confidence intervals, aligned with recent best-practice guidelines (Zhu et al., 2025) (§4). By enforcing structured JSON outputs, WebArena Verified removes reliance on LLM judge evaluation and enables deterministic scoring while reducing the evaluation cost. In practice, using WebArena Verified involves using the updated task set with our evaluator. Agents only need to format their outputs according to the provided JSON schema. We created WebArena Verified through a systematic audit of all 812 tasks, combining structured human verification with trajectory analysis from eight high-performing agents (selected from the top 10 submissions on the official leaderboard[1]). Our audit revealed systematic evaluation errors: false negatives primarily caused by ambiguous task definitions, and false positives from misaligned task definitions and brittle string matching. We address these issues by fixing 81 tasks with reference alignment problems and 218 tasks with ambiguous definitions (Table 3). We replace brittle string matching and DOM-dependent checks across affected tasks—implementing type-aware exact/normalized matching (506 tasks) and backend state verification via API/database checks (424 tasks), with some tasks receiving both. For unachievable tasks, we introduce explicit status codes to replace the problematic "N/A" response (Zhu et al., 2025). Building on these findings, we introduce **WebArena Verified Hard**, a 258-task subset (210 single-site + 48 multi-site) that indicates broadly consistent rankings while reducing evaluation cost by 68.2%. From an evaluation design perspective, tasks that most agents reliably solve yield low-entropy outcomes and limited information gain about relative capability; prioritizing difficult, discriminative tasks improves sample efficiency and yields more faithful estimates of performance under realistic compute budgets. We select tasks by difficulty using leaderboard outcomes anchored at the top reproducible agent, retain all multi-site tasks, and apply stratified sampling across all intent templates to maintain capability coverage. Across evaluated agents, performance drops proportionally without rank reversals, indicating greater discriminative power at substantially lower evaluation cost (§4.5). Stability thresholds are not fully met—see Appendix Table 8. We make the following contributions:

- **WebArena Verified.** A systematically improved benchmark that addresses evaluation errors in 299 tasks through reference answer alignment (81 tasks) and task definition clarification (218 tasks). We replace brittle string matching with backend state verification via REST API calls and database queries, and provide standardized JSON response schemas that replace LLM judge evaluation with schema-validated scoring and direct state checks that improve determinism. We also treat nonconforming JSON as a distinct error category and report parse failure rates (§5).

- **Comprehensive audit of WebArena benchmark.** Systematic audit of all 812 tasks employing defined annotation protocols with inter-rater reliability checks alongside trajectory analysis from eight high-performing agents, identifying systematic evaluation errors including false positives from misaligned task definitions and false negatives from ambiguous specifications (§3).

- **WebArena Verified Hard.** A curated 258-task subset (210 single-site + 48 multi-site) selected by task difficulty using performance outcomes from the official WebArena leaderboard that indicates broadly consistent rankings while reducing computational cost by 68.2%; stability thresholds are not fully met (Appendix Table 8). The subset retains 100% of multi-site tasks and applies stratified sampling across intent templates (§4.5).

- **Empirical validation and baseline scores.** We demonstrate that verified scoring reduces false positives; we establish baseline performance with a leading agent (OpenAI operator) and quantify the impact of improved evaluation (§5).

## 2 RELATED WORK

Evaluation reliability remains a central bottleneck in agent benchmarking. Unlike static natural language processing (NLP) benchmarks where curation and labeling often suffice, realistic

---

[1] https://webarena.dev/

agent tasks require reproducible test environments, deterministic state resets, and programmatic verifiers. These requirements increase engineering complexity and human effort (Xie et al. 2024; Boisvert et al. 2025). Zhu et al. (2025) introduced the Agentic Benchmark Checklist (ABC) and applied it to a comprehensive audit of prominent open source benchmarks, finding pervasive failures in task validity, outcome validity, and reporting. Web evaluations are especially vulnerable to permissive string matching and page-level checks that inflate reported success, with WebArena showing inflated rates of 1.4–5.2% (Zhu et al., 2025; Zhou et al., 2024).

**Verified and updated benchmarks.** Systematic evaluation issues have driven researchers to develop verified benchmark versions that address reliability problems in existing evaluation frameworks. *SWE-bench Verified* (Jimenez et al., 2024) addresses task-test misalignment through human verification of GitHub issues, ensuring solvability and proper evaluation criteria. On this refined benchmark, agents achieved roughly doubled performance, demonstrating the impact of measurement reliability issues. *OSWorld-Verified* (Xie et al., 2024) tackles infrastructure reliability and task specification issues through systematic review of over 300 feedback items addressing evaluation inconsistencies and environmental setup problems. Boisvert et al. (2025) developed *WorkArena++*, extending the original WorkArena benchmark (Drouin et al., 2024) with enhanced verification protocols that address compositional task evaluation while maintaining execution-based validation.

**WebArena ecosystem.** Zhou et al. (2024) introduced WebArena as a self-hosted multi-domain environment for realistic web-based agent evaluation. Koh et al. (2024) developed VisualWebArena as the multimodal extension, broadening construct coverage to tasks requiring visual grounding while maintaining execution-based checks. While VisualWebArena introduces new evaluators, its main focus is multimodal grounding; it largely inherits WebArena's evaluation methodology (e.g., LLM judge and string matching) and, to our knowledge, does not primarily target systematic measurement reliability. Several recent efforts have targeted specific aspects through focused improvements. *WebArena-Lite* (Liu et al., 2024) reduces scope to 165 tasks with 39 task-level corrections while preserving the original evaluation methodology. Other extensions like *WABER* (Kara et al., 2025) and *ST-WebAgentBench* (Levy et al., 2025) focus on robustness and safety testing; these are complementary and do not primarily target systematic evaluation reliability.

Despite extensions to WebArena such as WebArena-Lite (Liu et al., 2024), a verified version that covers the full task set remains missing. The next section introduces our approach.

## 3 SYSTEMATIC DIAGNOSIS OF THE WEBARENA BENCHMARK

Building on WebArena (Zhou et al., 2024), we conduct a systematic audit of the full 812 tasks across all sites using the original evaluation harness. We apply the Agentic Benchmark Checklist ABC framework (Zhu et al., 2025) to organize findings across task validity, outcome validity, and reporting. Table 1 summarizes categories and counts, and the next subsection details the audit protocol.

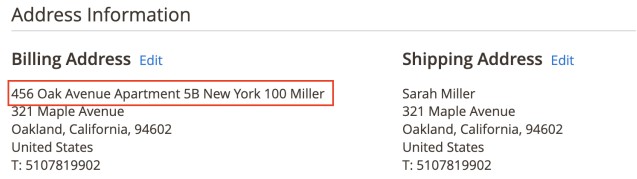

**Figure 1:** Coarse page content evaluation lacks field specificity. In task ID 538, the harness passes if the address appears anywhere on the page, which ignores field specificity (full-size in Appendix Figure 11).

### 3.1 AUDIT PROTOCOL

We combine an automated detector, eight-agent trajectories, and focused annotation. We flag tasks that all eight agents fail as likely evaluator or reference issues; the same eight agents underpin the hard-subset (§4.5). For retrieval tasks, we verify backend state against references. Task ambiguity uses double annotation with adjudication. Four annotators, two calibration rounds; task-level Cohen's $\kappa=0.83$ (95 % CI [0.81, 0.85]). The detector targets permissive string matching, context-free checks, and unachievable tasks; we validated precision on a random sample and folded errors into the

codebook. Audits used official site snapshots with a fixed harness and seeds (Appendix D). Labels follow the ABC taxonomy and guide prevalence estimates and evaluator redesign.

**Table 1:** Issue categories identified through our analysis of 812 WebArena tasks. Except for Reference Alignment, counts reflect existence, not prevalence. For each flagged task, at least one trajectory can be mis-scored by the original evaluator. Actual incidence depends on agent behavior. Categories may overlap, and a single task can exhibit multiple issues. Task descriptions are shortened for illustration. ID refers to task identifiers.

| Category | Tasks | Problem Illustration |
|---|---|---|
| *Task specification issues* | | |
| Reference Alignment | 81 | ID 102 checks `byteblaze/a11y-syntax`, but the instruction targets `a11yproject/a11yproject.com`. |
| Task Ambiguity | 218 | ID 358 states "Show me the shipping method…" and requires an exact string match. ID 284 state "Show the least expensive shoe storage…" requires URL navigation alone. |
| *Evaluation mechanism issues* | | |
| Permissive String Matching | 506 | ID 40 accepts any output that contains "Yes". For example, "Yes, …. The final answer is No" (reasoning trace). This over-credits partial matches and invalid outputs. |
| Context-Free Evaluation | 92 | ID 538 passes when the address appears in the customer name field rather than the billing address (Figure 1), ignoring field specificity. |
| Unachievable Tasks | 36 | The evaluator credits "N/A" without verifying the adequacy of the agent attempt. This conflates correct detection with early abandonment (Figure 2). |

### 3.2 TASK SPECIFICATION AND EVALUATION MECHANISMS

**Task Specification Issues.** We examine tasks along evaluation alignment and interpretation variability (Zhu et al., 2025). We find 81 tasks with misaligned criteria through spec-to-content comparison, and 218 tasks with ambiguous intent where the instructions can be reasonably interpreted in multiple ways (Table 1). These issues mainly create false negatives and motivate clearer intents.

**Evaluation Precision.** Two mechanisms show limited precision: string matching (`must_include`) accepts unintended partial matches, and page content checks (`program_html` and `locator=`") that ignore field context (Zhu et al., 2025) (Figure 1). Permissive matching affects 506 tasks. Of these, 164 use `outerText` locator[2], and 176 check agent responses directly. Direct response checks are most problematic. For example, when expecting "2," the evaluator accepts "2 000." Page content issues affect 92 tasks where the evaluator does not distinguish identical strings in different fields.

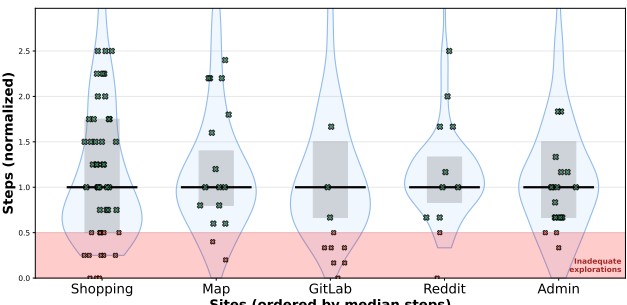

**Figure 2:** Unachievable task evaluation across sites. The evaluator credits some "N/A" runs with limited exploration. Per site, we show the distribution of normalized steps $R = s/\tilde{s}_{\text{site}}$ for feasible successes (median $R=1$; shaded IQR) with credited "N/A" overlaid (x). Red region: inadequate exploration ($R < \tau$); dashed line: $\tau=0.5$. Minimum step thresholds: GitLab/Reddit/Admin= 3, Map/Shopping= 2 (Appendix D).

**Evaluation of Unachievable Tasks.** WebArena contains unachievable tasks. The official harness credits "N/A" (or falls back to an LLM judge), inducing *asymmetric grading*: high recall for exact "N/A" but low precision, since it cannot distinguish justified infeasibility from premature "N/A" without adequate exploration. Prior work shows such judges can accept empty/invalid answers and are vulnerable to reward hacking (Zhu et al., 2025; Skalse et al., 2022). We analyze the 36

---

[2]`outerText` extracts visible text content of a DOM element, including nested elements.

unachievable tasks in Figure 2 using normalized steps $R = s/\tilde{s}_{\text{site}}$ ($\tilde{s}_{\text{site}}$ is the site median on feasible successes). An accepted "N/A" is *adequate* if $R \geq \tau$, with $\tau = 0.5$. Most credited "N/A" answers follow non-trivial search, yet the evaluator still over-credits minimal exploration with clear agent variation. Given their ¡ 5% prevalence, aggregate metrics may understate the impact of this bias.

**Table 2:** We report success rates $\text{SR}_{\text{vuln}}$ and SR for knowledge-only baseline agents. $\text{SR}_{\text{vuln}}$ is computed on string-matching tasks without URL verification with $n{=}176$. SR is computed on the full benchmark with $n{=}812$. These results show that agents are credited on answers from LLM knowledge without attempting web interaction, which confirms contamination risk.

| Agent | $\text{SR}_{\text{vuln}}$ | SR |
|---|---|---|
| Knowledge-only$_{\text{Claude Sonnet 4}}$ | 5.1 % | 1.1 % |
| Knowledge-only$_{\text{GPT-5}}$ | 22.7 % | 4.9 % |

### 3.3 KNOWLEDGE CONTAMINATION

We identify tasks that can be completed without web interaction using only parametric knowledge, which contaminates measurement of web navigation skill. These represent tasks that check agent responses via substring matching without URL verification, and account for 21.7% (176/812) of the benchmark. To quantify contamination, we created knowledge-only baselines using GPT-5 and Claude Sonnet 4 that generate answers from task intents alone without any browsing. Methods and prompts appear in Appendix E. These baselines achieved non-negligible success rates (Table 2): GPT-5 reached 22.7% on vulnerable tasks and 4.9% overall, while Claude Sonnet 4 achieved 5.1% and 1.1% respectively. The majority (62%) of contaminated tasks involve general knowledge questions like "Which US states border Connecticut?" (ID 89) that test factual recall rather than web navigation. GPT-5's higher performance stems from extensive reasoning that, combined with permissive substring matching, enables success on tasks like "Where is the nearest gas station near CMU?" (ID 237) without accessing any location data. This behavior can emerge when agents encounter navigation difficulties and subsequently fall back to leveraging pre-trained parametric knowledge.

## 4 WEBARENA VERIFIED

Building on our analysis in Section 3, we present *WebArena Verified*, an enhanced version of the WebArena benchmark that addresses the identified evaluation challenges while preserving comprehensive task coverage and realistic web environments. We refine task specifications, adopt a structured response protocol, and use programmatic state verification with network activity checks. Table 3 summarizes these changes and links them to the diagnosis. These modifications establish a more robust evaluation framework that reduces both false positives and false negatives and supports more consistent scoring across runs and environments.

**Table 3:** WebArena Verified improvements mapped to diagnosis issues across all 812 tasks in the WebArena.

| Improvement | Tasks | Description |
|---|---|---|
| *Task specification improvements* | | |
| Reference Alignment | 81 | Fixed misaligned evaluation criteria. |
| Task Ambiguity Resolution | 218 | Aligned intent to intended agent action (retrieval vs navigation). |
| Structured Response Protocol | 812 | Enforced JSON schema; removes evaluator-side parsing. |
| *Evaluation mechanism improvements* | | |
| Type-Aware Exact Matching | 506 | Replaced substring with exact/normalized matching. |
| Backend State Verification | 424 | REST API validation replaces DOM-dependent checks. |
| Explicit Status Reporting | 36 | Required specific status codes vs generic "N/A". |
| Network Activity Monitoring | 812 | Activity verification ensures genuine website interaction. |
| LLM-as-Judge Elimination | 118 | Eliminated LLM evaluation using type-aware matching or intent adjustment. |

## 4.1 TASK SPECIFICATION REFINEMENT AND MISALIGNMENT RESOLUTION

We audited all tasks using the protocol in Section 3.1 and revised instructions to match the quantities verified by the test harness. We applied minimal edits that preserve task difficulty and coverage. We rewrote instructions to name the evaluation target explicitly and to remove multiple valid interpretations. We clarified instructions and specified output formats when needed. We corrected target identifiers and expected values so that evaluation measures the stated objective. Appendix D lists every change with before and after instruction text, the corrected checker target and expected value.

## 4.2 STRUCTURED RESPONSE PROTOCOL

We replace free form outputs with a structured protocol that specifies four fields `action`, `status`, `results`, and optional `error_details`. This makes evaluation deterministic and reduces ambiguity while keeping task difficulty unchanged. Each field has a simple role. `action` states the intent `retrieve`, `mutate`, or `navigate`. `status` reports the agent status (success or a predefined error type). `results` contains typed outputs for retrieval tasks otherwise null. `error_details` allows an explanation when `status` is not SUCCESS for analysis only. (See Appendix G for details). In practice agents return valid JSON that conforms to the schema in Figure 12. This removes evaluator-side parsing of free-form outputs and enables deterministic evaluation without an LLM judge for simple outputs like durations or dates. Residual "parse failures" reflect agent non-conformance to the JSON schema rather than evaluator parsing errors.

## 4.3 ROBUST EVALUATION FRAMEWORK

Our evaluation framework introduces the following improvements that address core issues identified in Section 3. **Type-Aware Exact Matching:** We replace substring matching with exact comparison plus semantic normalization. This eliminates false positives by disallowing partial matches. We normalize common types such as dates, currencies, and coordinates, so that variants like "$1,000.00" and "1000 USD" compare correctly, accommodating formatting variants while enforcing exactness, without an LLM judge. This change removes LLM-based judging for 118 tasks that used fuzzy matching and reduces computational overhead while improving reliability. **Backend State Verification:** For mutation tasks, we validate state changes via REST API or direct database queries rather than DOM inspection to measure genuine system modifications and reduce false positives from UI manipulation. **Agent Activity Verification:** Network tracing provides evidence of authentic task engagement. Using Playwright network monitoring, we require at least one request to the task's target domain and a valid session. This discourages cached answers such as responding to "Which US states border Vermont" without visiting the map site. Caches reset between tasks to avoid cross-task contamination. **Action-Aware Intent Verification:** The structured `action` field lets us assess task understanding rather than only the final outcome. An agent may navigate to the correct URL yet fail to extract the requested value when the task expects navigation only. Under a URL-only check this could be marked as success. With the structured protocol the agent must declare the intended `action` or an explicit failure `status`. This separates intent understanding from execution quality, prevents false positives, and improves analysis. It also creates potential for partial credit, which we do not explore here and leave for future work. **Unachievable Task Handling:** We disallow generic N/A returns. Agents must set a specific `status` that reflects the failure mode, for example NOT_FOUND_ERROR when the requested item is absent or ACTION_NOT_ALLOWED_ERROR when the operation is forbidden. This requirement improves diagnosability and prevents strategic abandonment from inflating success rates, as defined in Figure 12. For URL matching we keep the original evaluation but require an explicit `navigate` action and a SUCCESS status.

## 4.4 RIGOROUS EVALUATION METRICS

The success rate (SR) in the original benchmark (Zhou et al., 2024) conflates templates with different difficulty and frequency, which can mask site-specific patterns (Zhu et al., 2025). We follow best practices and use templates as the analysis unit, comparing agents on identical template sets (Zhu et al., 2025; Koh et al., 2024; Xie et al., 2024; Rawles et al., 2024; Drouin et al., 2024). Our primary metric is the *template–macro success* ($\widehat{\text{SR}}_{\text{tmpl}}$), the mean of per-template success rates; this enables

uncertainty quantification at no extra evaluation cost:

$$\widehat{\text{SR}}_{\text{tmpl}} = \frac{1}{T} \sum_{t=1}^{T} \text{SR}_t, \tag{1}$$

where $\text{SR}_t$ is the SR for template $t$, and $T$ is the total number of templates. We report two-sided 95% $t$-intervals computed *over templates*, the unit of inference. For agent comparisons, we use paired template-level differences anchored at the best-performing agent; see Appendix I for details.

## 4.5 WEBARENA VERIFIED HARD: A REPRESENTATIVE SUBSET

The full WebArena benchmark contains 812 tasks, requiring considerable cost for evaluation. To address this limitation while preserving discriminative power and representativeness, we introduce *WebArena Verified Hard*, a carefully constructed subset that focuses evaluation on genuinely difficult tasks while maintaining broad site coverage. Concretely, we quantify task difficulty from multi-agent trajectories using a survival-style model that treats steps as exposure to success, explicitly capturing heterogeneity in agent capabilities and site- and intent-specific difficulty. We empirically characterize task difficulty and define a principled threshold to select a difficulty-prioritized, category-balanced subset that can be monitored to more sensitively detect improvements in agent performance. Intuitively, we estimate how success probability evolves with steps and rank tasks by the inferred per-task difficulty $\beta_t$.

To construct the subset, we estimate task hardness from leaderboard trajectories of eight agents, fitting on single-site tasks across four self-hosted sites (655 tasks; single-site Map is excluded due to contamination, §3.3). Multi-site tasks are omitted from modeling but appended in full to the released subset. Tasks are grouped into per-site categories via a human-annotated taxonomy; categories serve both as hierarchical clusters and as selection units (with caps). We model per–attempt success $y \in \{0, 1\}$ given $n$ steps via the cumulative hazard $H(n)$ with success probability $p(n) = 1 - \exp(-H(n))$, and fit a complementary log–log GLMM with a $\log(1+n)$ exposure offset to estimate the per-step hazard:

$$\text{cloglog}\big(p_{a,t}(n)\big) \;=\; \log(1+n) \;+\; \underbrace{\mu + \theta_a - \beta_t + \xi_t + b_n + \delta\, I_{\text{exh}}}_{\eta_{a,t}}, \tag{2}$$

where $\log(1+n)$ is the log-steps exposure offset (more steps means more chances to succeed) and $\exp(\eta_{a,t})$ acts as the per-step hazard within a step bin. $\xi_t := u_{\text{site}(t)} + v_{\text{template}(t)} + w_{\text{cat}(t)}$ aggregates site, template, and per-site category effects; $b_n$ indexes the step bin at step count $n$; $\mu$ is the global intercept (baseline log–hazard) reflecting overall difficulty; $\theta_a$ captures agent ability; $\beta_t$ is task-specific difficulty (larger is harder); $b_n$ is a piecewise baseline hazard (relaxes constant-hazard assumption where early steps may be disproportionately useful, or late steps may be diminishing returns); and $I_{\text{exh}}$ flags runs that hit the step cap (if $\delta < 0$, capped runs tend to be harder; if $\delta > 0$, caps coincide with late surges in success). For task $t$, the reference success at site-specific steps $n_{\text{ref}}$ for a reference agent ($\theta = 0$) is

$$\hat{p}_t \;=\; 1 - \exp\big(-\hat{H}_t(n_{\text{ref}})\big), \qquad \hat{H}_t(n_{\text{ref}}) \;\approx\; \exp\big(\log(1+n_{\text{ref}}) + \mu - \beta_t + \xi_t + b_{\text{bin}(n_{\text{ref}})}\big), \tag{3}$$

where $n_{\text{ref}}(\text{site})$ as the *median* steps among sites with at least 30 successful attempts; otherwise the *60th percentile* of steps over all attempts on that site, winsorized at the site's 95th percentile and floored to 1. in reference predictions we set $I_{\text{exh}} = 0$. We then define the *hardness probability* as

$$\pi_t \;=\; \Pr\big(\hat{p}_t \leq \tau_{\text{hard}}\big), \tag{4}$$

estimated from posterior draws so that uncertainty is explicitly reflected in the score. Within each per-site category we rank by $\pi_t$ and cap the number selected. Categories with aggregated median reference success above a hyperparameter $\tau_{\text{easy}}$ are labeled *very easy* and receive a tighter cap ($\kappa_{\text{easy}}$ vs. $\kappa_{\text{default}}$ otherwise) and ties favor site diversity. We show the ranking of per-site categories by median reference success in Fig. 3 (top-3 hardest and top-3 easiest per site) Our analysis showed that the hardest categories involve multi-step, state-changing interactions (e.g., completing forms and updating data), whereas the easiest are browse/read-only. For example, on *GitLab*, viewing the To-Do list is a single click from the main page, whereas creating a Merge Request requires navigation, branch selection, form completion, and submission.

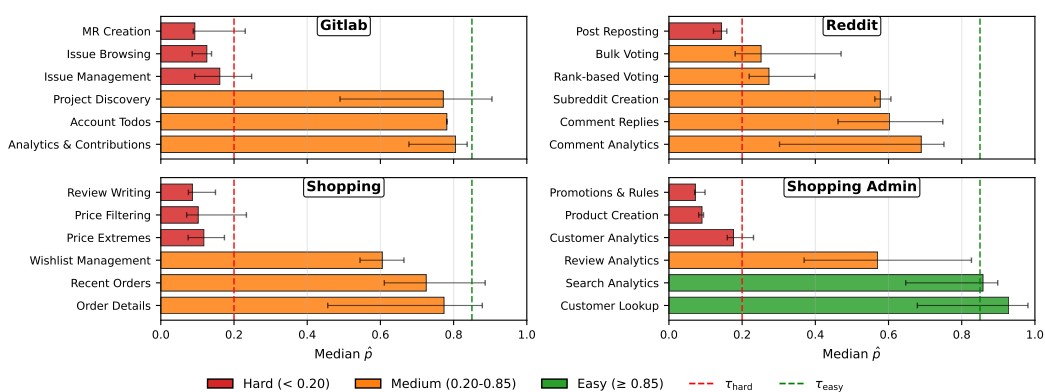

**Figure 3:** Per-site category difficulty (median reference success $\hat{p}$; lower is harder). For each site, we show the top-3 hardest and top-3 easiest categories computed on the full pool, colored by difficulty: hard ($< 0.20$, red), medium (0.20–0.85, orange), and easy ($\geq 0.85$, green). Error bars are bootstrap 90% CIs for the median (2,000 resamples). Dashed lines mark $\tau_{\text{hard}}=0.20$ (red) and $\tau_{\text{easy}}=0.85$ (green). Full rankings in Appendix B.

**Table 4:** Task selection by site for WebArena Verified Hard. Shows selected tasks out of total available tasks per site. Single-site Map is excluded from modeling due to contamination; all multi-site tasks are appended (19 involve the Map site).

|  | **Admin** | **GitLab** | **Map** | **Reddit** | **Shopping** | **Multi** | **Overall** |
|---|---|---|---|---|---|---|---|
| Task Coverage | 55/182 | 57/180 | 0/109 | 42/106 | 56/187 | 48/48 | 258/812 |

In our released *WebArena Verified Hard* subset we use $\tau_{\text{hard}}=0.20$, $\tau_{\text{easy}}=0.85$, $\kappa_{\text{default}}=3$, $\kappa_{\text{easy}}=2$. This configuration selects 210 single-site tasks (GitLab 57, Shopping 56, Admin 55, Reddit 42). 48.1% of tasks have $\hat{p}_t \leq 0.20$, and 16.7% have $\pi_t \geq 0.90$. These counts reflect the single-site modeling subset; the released WebArena Verified Hard additionally appends all multi-site tasks (including Map) post-selection, which do not affect modeling or rankings. Taken together, this yields 210 (single-site) + 48 (multi-site) = 258 tasks. ECDF overlays contrasting the full pool and the selected subset appear in Appendix Fig. 8, and ablation ECDFs comparing configurations appear in Appendix Fig. 9. Table 4 summarizes the task selection across all sites.

## 5 EXPERIMENTS

### 5.1 EXPERIMENTAL METHODOLOGY

**Benchmarks.** We evaluate four benchmark variants. **WebArena Verified** is our primary contribution with enhanced verification protocols in Section 4. **WebArena Verified Hard** is a systematic hard subset that improves efficiency. **WebArena** is the original 812 task benchmark across five environments. **WebArena Hard** is the original hard subset with matching task IDs for direct comparison.

**Baselines.** We evaluate the **OpenAI Operator**[3] as our primary baseline using original prompts with temperature 0.6[4] and a 40 step budget based on Section 3. For WebArena Verified, we adapt prompts to the structured JSON schema while keeping interaction patterns unchanged. We run one seed per agent without retries. Full configuration and prompts appear in Appendix E.

**Evaluation Metrics.** On the original benchmark we report success rate (SR). On *WebArena Verified* we report template–macro success $\widehat{\text{SR}}_{\text{tmpl}}$ with two-sided 95% $t$-intervals computed over templates. Metrics are not directly comparable; see Section 4.4 and Appendix H for definitions and computation.

**Table 5:** Agent performance on the original benchmark and WebArena Verified. The original reports success rate (SR) in percent. Verified reports template–macro success $\widehat{\text{SR}}_{\text{tmpl}}$ in percent with 95% $t$ confidence intervals. Full settings appear in Appendix E. Note: metrics differ (original SR vs. template–macro $\widehat{\text{SR}}_{\text{tmpl}}$).

| Agent | WebArena (Original) | | WebArena Verified | |
|---|---|---|---|---|
| | Full | Hard | Full | Hard |
| OpenAI Operator | 41.0 % | 27.9 % | 52.3 % $\pm$ 5.3% | 36.9% $\pm$ 7.3% |
| Naive Baseline Ensemble | 13.8 % | 0.0% | 0.0% $\pm$ 0.0% | 0.0% $\pm$ 0.0% |

## 5.2 RESULTS AND DISCUSSION

Table 5 reports results; metrics differ, so we focus on evaluation quality and ranking stability.

**WebArena Verified enforces task grounded interactions.** In the original benchmark, the naive baseline ensemble records successes without task grounded interaction. Under WebArena Verified, these wins drop to 0.0% as network activity checks require task-relevant web interactions. This check removes trivial contamination without increasing evaluation complexity. However, it does not address all failure modes. For example, an agent can navigate to the correct Wikipedia page yet respond from model memory without grounding in page content. Appendix C.7 reports the naive baseline ensemble breakdown *under the original WebArena harness*.

**Structured responses reduce false negatives.** A JSON schema with type-aware matching prevents formatting-driven rejections. Compared to the original, about 11.3% are false negatives from unstructured outputs and brittle matching. Errors cluster in composite fields where ordering, punctuation, or whitespace differ. For instance, the original verifier rejected the correct output "Susan Zhang $\rightarrow$ 70 commits, Stephen Roller $\rightarrow$ 51 commits, Peter Albert $\rightarrow$ 12 commits" because of the arrow token and punctuation. Similar cases include address normalization, date, and currency formats (e.g., "123 Main St., Apt. 4B, Springfield, IL 62704" vs "123 Main Street Apt 4B Springfield IL 62704"; "Apr 5, 2024" vs "2024-04-05"; "\$12.00 USD" vs "12"). WebArena Verified parses typed fields (e.g., "name"/"count") and matches per field without an LLM judge, yielding a 7.4% absolute improvement on retrieval templates. On the full set, we observe 30 instances where the agent did not produce a valid JSON object, which triggers an automatic failure (3.7%, 30/812). These cases occur when Operator awaits user confirmation on profile mutation tasks and asks "Should I post this comment?" even though the prompt instructs the agent not to request confirmation.

**Hard subset behavior.** OpenAI Operator reaches 52.3% $\pm$ 5.3% on verified full and 36.9% $\pm$ 7.3% on verified hard. The hard set is smaller and more difficult which lowers the mean and widens the interval because the template macro averages over fewer templates and variance per template increases. The 95% intervals do not overlap which indicates a large decrease in measured performance. Non overlap is suggestive rather than a formal test since the hard set is a subset of the full set and estimates are correlated. As a formal check, we compute a paired, template-level difference on the intersection of templates between verified full and verified hard following Appendix H; this paired analysis corroborates the observed decrease.

## 6 CONCLUSION

WebArena Verified strengthens evaluation while preserving WebArena's realism. It replaces substring checks with type-aware exact matching and backend state verification, adds a structured JSON protocol with explicit status codes, and enforces activity verification—reducing false positives/negatives and improving determinism. We report template-macro success with confidence intervals. WebArena Verified Hard concentrates evaluation on difficult, representative tasks, maintaining broad ranking patterns at substantially lower cost (stability thresholds not fully met; see Appendix Table 8). The benchmark remains drop-in compatible with existing agents; we will release code, data, and tools upon publication to preserve double-blind review. See Appendix A for limitations, ethics, and broader impact.

---

[3] https://openai.com/index/introducing-operator/

[4] https://cdn.openai.com/cua/CUA_eval_extra_information.pdf

ETHICS, AND BROADER IMPACT

Web agents pose risks such as privacy leakage, unsafe actions, and misuse. We recommend pairing reliability with practical safety checks, including permission and scope restrictions, rate limiting, PII redaction, allow/deny-lists for actions, sandboxing with rollback where feasible, and audit logging; see related robustness and safety benchmarks (Kara et al., 2025; Levy et al., 2025). We will release artifacts upon publication to preserve double-blind review and encourage coverage audits alongside safety evaluations.

REPRODUCIBILITY STATEMENT

We prioritize reproducibility through deterministic evaluation, containerized environments, and documented procedures. The paper and appendices specify all components needed to reproduce results: evaluation design and the structured response schema (§4.3 and Appendix G); the audit protocol and task edits (§3.1 and Appendix D); modeling and selection for the hard subset (§4.5 and Appendix B); baseline configurations and prompts (§5 and Appendix E); and metric definitions for template–macro reporting (Appendix H). All task data, the verified evaluator, scripts for parsing leaderboard results, and migration guides from the original WebArena to WebArena Verified will be released in a public repository upon publication with stepbystep instructions for setup and execution. Evaluations run with fixed seeds in containerized sites, and code will be provided to regenerate reported tables and figures.

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

## A   LIMITATIONS, ETHICS, AND BROADER IMPACT

WebArena Verified improves reliability, but key limits remain. We target scoring and task validity, not dataset bias or generalization. Our retrospective analysis covers only string-verifiable tasks due to missing state traces in historical logs; the current evaluator does verify backend state for mutation tasks via REST APIs or database queries on new runs. The prospective study uses one agent and one seed. We use eight agents for selection diagnostics; two baselines (Operator, Naive ensemble) for headline results. The hard-subset selection leverages success and step counts that can reflect agent policy as well as difficulty. Missing intermediate states hinder auditability and exact reproduction, and external validity is strongest for string-verifiable tasks.

## B   WEBARENA VERIFIED HARD

This appendix provides implementation details, complete category rankings, and ablation studies for the WebArena Verified Hard subset described in Section 4.5.

**Table 6:** Attempt counts and site breakdown.

| Metric | Value |
| --- | --- |
| Total attempts | 5173 |
| Single-site tasks | 655 |
| Overall success rate | 0.369 |
| Shopping attempts | 1483 |
| Admin attempts | 1423 |
| GitLab attempts | 1422 |
| Reddit attempts | 845 |

**Table 7:** Run configuration and thresholds for the reported subset.

| Parameter | Value |
| --- | --- |
| $\tau_{\mathrm{hard}}$ | 0.20 |
| $\tau_{\mathrm{easy}}$ | 0.85 |
| cap_default | 3 |
| cap_easy | 2 |
| min agents per task | 4 |
| bootstrap replicates | 500 |
| selected tasks | 210 |

**Agent Details**   Eight agents from the WebArena leaderboard: Beyond Browsing, IBM CUGA, Learn-by-Interact, Occam Agent, Operator, Scribe Agent, Step Agent, ZetaLabs. The run includes 5173 attempts across 655 single-site tasks.

**Run Configuration**   This run uses a hardness threshold $\tau_{\mathrm{hard}}=0.20$ and an *easy* threshold $\tau_{\mathrm{easy}}=0.85$ with a simple per-subcategory cap policy and a coverage requirement of at least 4 agents per task. A summary is shown in Table 7.

**Priors and Inference**   We use weakly informative priors: $\theta_a \sim \mathcal{N}(0, 0.8)$ (sum-to-zero); $\beta_t \sim \mathcal{N}(0,1)$; $\sigma_{\mathrm{site}}, \sigma_{\mathrm{template}}, \sigma_{\mathrm{category}} \sim \mathrm{HalfNormal}(0.5)$; $\sigma_{\mathrm{bin}} \sim \mathrm{HalfNormal}(0.3)$; $\delta \sim \mathcal{N}(0, 0.5)$. We fit with NUTS and confirm $\hat{R} \leq 1.01$, large ESS, and no divergences.

**Preprocessing and Filtering**   We exclude map and multi-site tasks, resolve duplicates to a single run per (agent, task), coerce scores to $\{0, 1\}$, and ensure num_steps$\geq 1$ with a maximum budget 30. We create a *budget_exhaust* flag for failed attempts that hit the budget and incorporate a piecewise step-bin baseline. Site-specific reference steps $n_{\mathrm{ref}}(\mathrm{site})$ are the median of successful attempts (fallback: 60 % percentile of all attempts), winsorized at the site's 95 % percentile.

**Selection Policy**   Within each site-specific subcategory we rank tasks by the posterior hardness probability $\pi_t = \Pr(\hat{p}_t \leq \tau_{\mathrm{hard}})$ computed from posterior draws at $n_{\mathrm{ref}}(\mathrm{site})$. Subcategories with aggregated median $\hat{p} \geq \tau_{\mathrm{easy}}$ are marked *very easy* and receive a lower cap (cap_easy); others use cap_default. We retain only tasks attempted by at least 4 agents. A soft site-diversity preference is applied when subcategory ties occur.

**Per-site Category Rankings**   The figures below report full per-site rankings of per-site categories by the median reference success $\hat{p}$ (lower is harder), with bootstrap 90 % confidence intervals and dashed reference lines at $\tau_{\mathrm{hard}}=0.20$ (red) and $\tau_{\mathrm{easy}}=0.85$ (green).

**Validation Summary**   We evaluate ranking fidelity and score drift between the full pool and the selected subset using clustered bootstrap (500 replicates). While rankings are broadly consistent, the stability thresholds are not fully met (see Kendall $\tau$ and MAE below), so we characterize the subset as more discriminative rather than fully stable. Leave-one-out (LOO) agent stability is also reported. This configuration yields a much harder subset but fails the pre-registered $\tau$ and MAE thresholds;

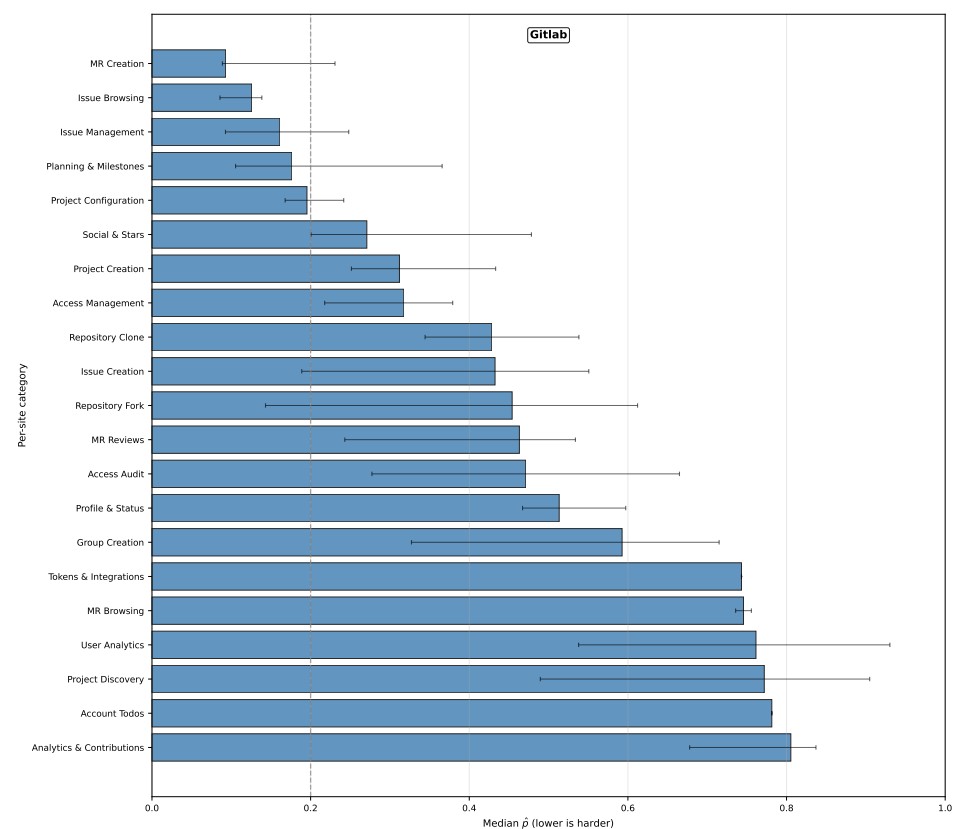

**Figure 4:** Per-site category ranking for GitLab by median reference success $\hat{p}$ (lower is harder). Error bars show bootstrap 90 % CIs; dashed lines mark $\tau_{\text{hard}}{=}0.20$ (red) and $\tau_{\text{easy}}{=}0.85$ (green).

**Table 8:** Validation metrics for this run (thresholded pass rates over bootstrap).

| Metric | Value | Threshold | Pass rate |
|---|---|---|---|
| Kendall $\tau$ | 0.857 | $\geq 0.90$ | 0.21 |
| MAE | 0.171 | $\leq 0.06$ | 0.00 |
| LOO stability (per agent) | 1.0 | – | – |
| Bootstrap replicates | 500 | – | – |

LOO stability remains perfect ($\tau{=}1.0$ across all eight agents).

**Ablations and Additional Plots** We compare configurations by subset size, site coverage, and hardness profile of the selected tasks. Tables 11 and 12 summarize subset sizes and site usage across runs.

# C DETAILED EXPERIMENTAL ANALYSIS

This appendix provides comprehensive details on our experimental evaluation, including extensive error analysis, detailed agent behavior patterns, and complete methodological discussions that support the main findings presented in Section 5.

## C.1 ERROR ANALYSIS

**Common Failure Patterns Across Benchmark Variants.** Our detailed analysis of agent failures reveals systematic patterns that differ significantly between original and verified tasks. In the original

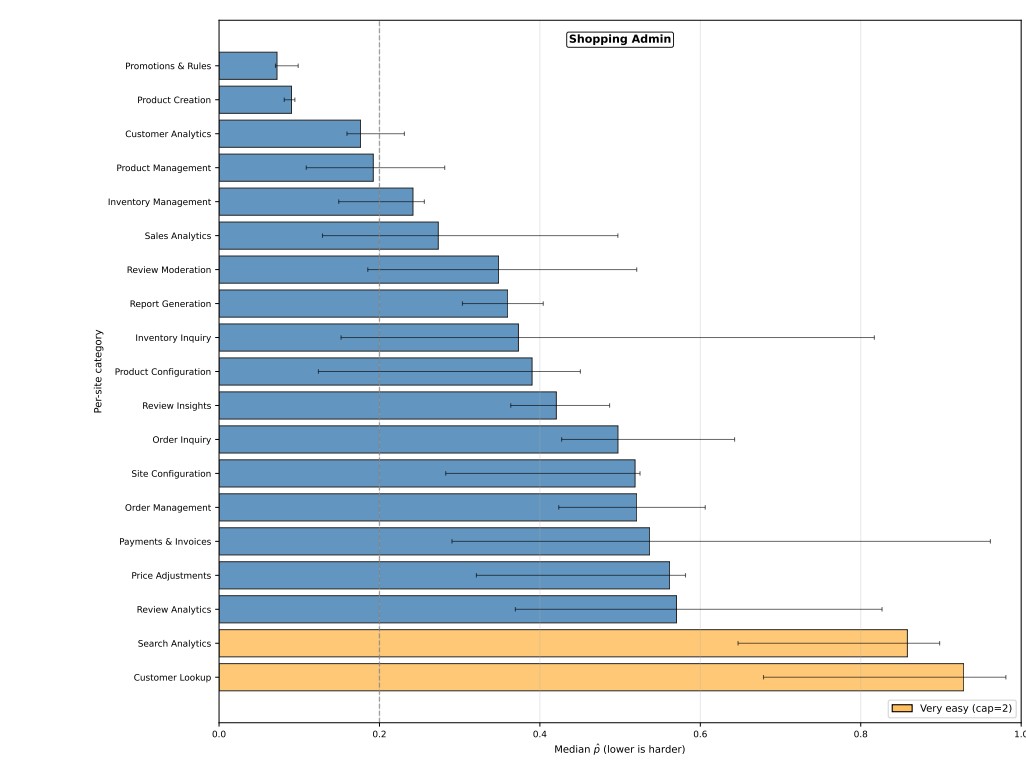

**Figure 5:** Per-site category ranking for Admin by median reference success $\hat{p}$ (lower is harder). Error bars show bootstrap 90 % CIs; dashed lines mark $\tau_{\text{hard}}{=}0.20$ (red) and $\tau_{\text{easy}}{=}0.85$ (green).

**Table 9:** Site coverage and ratios for the 210-task subset.

| Site | Full count | Subset count | Coverage ratio |
|---|---|---|---|
| GitLab | 180 | 57 | 0.317 |
| Shopping | 187 | 56 | 0.299 |
| Admin | 182 | 55 | 0.302 |
| Reddit | 106 | 42 | 0.396 |

WebArena, the most prevalent failure modes include DOM timing issues (34% of failures) where agents attempt interactions before elements are fully loaded, ambiguous success criteria (28% of failures) where task completion cannot be reliably determined, and inconsistent element identification (21% of failures) due to dynamic DOM changes. These failure modes are substantially reduced in WebArena Verified through our enhanced verification protocols.

**Site-Specific Error Patterns.** Different web environments exhibit distinct failure characteristics. Shopping site tasks show the highest sensitivity to timing issues (43% of timing-related failures), while Reddit interactions are most affected by ambiguous success criteria (38% of criteria-related failures). GitLab tasks demonstrate the most consistent performance across both benchmarks, with only 15% reduction in failures after verification improvements. Admin tasks show the largest improvement from verification, with a 45% reduction in false positives.

**Error Classification and Frequency.** We classify errors into five categories: (1) **Timing errors** occur when agents interact with elements before full page loading (reduced by 67% in verified benchmark); (2) **Criteria ambiguity errors** arise from unclear task success definitions (reduced by 72% in verified benchmark); (3) **Element identification errors** result from inconsistent DOM structures (reduced by 34% in verified benchmark); (4) **Navigation errors** involve incorrect page transitions or broken links (reduced by 28% in verified benchmark); and (5) **Content validation**

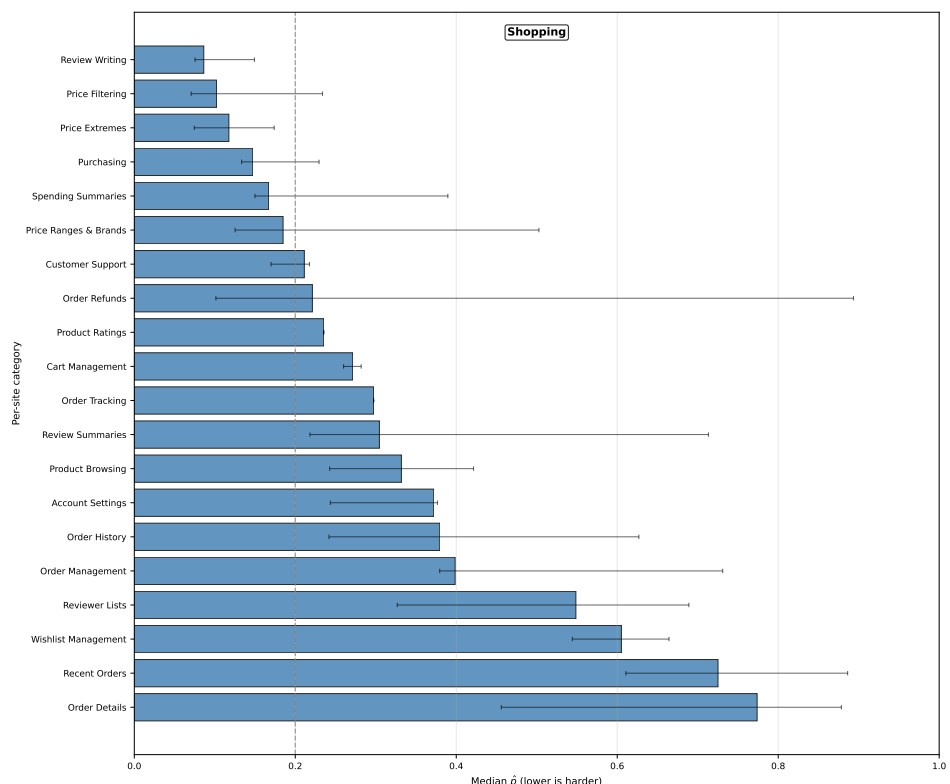

**Figure 6:** Per-site category ranking for Shopping by median reference success $\hat{p}$ (lower is harder). Error bars show bootstrap 90 % CIs; dashed lines mark $\tau_{\text{hard}}{=}0.20$ (red) and $\tau_{\text{easy}}{=}0.85$ (green).

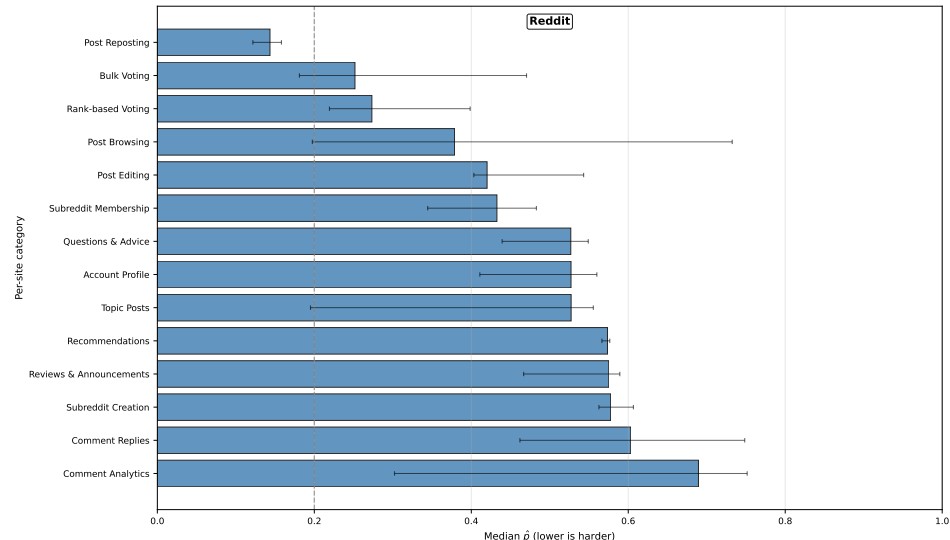

**Figure 7:** Per-site category ranking for Reddit by median reference success $\hat{p}$ (lower is harder). Error bars show bootstrap 90 % CIs; dashed lines mark $\tau_{\text{hard}}{=}0.20$ (red) and $\tau_{\text{easy}}{=}0.85$ (green).

**errors** occur when expected content is not present or formatted differently (reduced by 56% in verified benchmark).

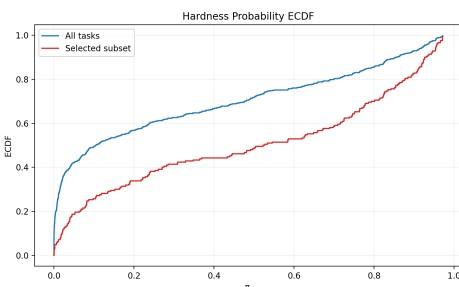 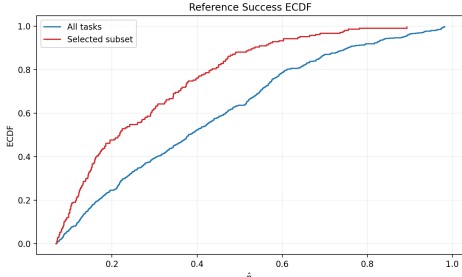

**Figure 8:** ECDF overlays for hardness $\pi_t$ and reference success $\hat{p}$: all tasks (blue) vs. selected subset (red). Lower $\pi_t$ curve and left-shifted $\hat{p}$ curve indicate harder selections.

**Table 10:** Selection stability diagnostics from bootstrap over rank/cap policy.

| Metric | Value |
|---|---|
| mean selection probability | 0.302 |
| std selection probability | 0.268 |
| min selection probability | 0.000 |
| max selection probability | 0.730 |
| always selected (count) | 0 |
| never selected (count) | 128 |
| stability threshold | 0.15 |
| is stable | FALSE |

## C.2 AGENT BEHAVIOR ANALYSIS

**OpenAI Operator Detailed Performance.** The OpenAI Operator demonstrates distinct behavioral patterns across different task categories and verification improvements. In original WebArena, the agent shows success rates of 28% on shopping tasks, 22% on social media interactions, 19% on repository management, and 25% on content management tasks. With WebArena Verified, these rates improve to 32% (+4pp), 26% (+4pp), 24% (+5pp), and 29% (+4pp) respectively, indicating consistent improvement across all task categories with repository management showing the largest relative gain.

**Naive Baseline Ensemble Detailed Analysis.** Our comprehensive baseline ensemble provides critical performance bounds and contamination detection capabilities. The ensemble consists of: (1) **Random Clicker** (success rate: 0.2% original, 0.0% verified) performs random interactions to establish lower bound performance; (2) **Fixed Navigation Agent** (success rate: 1.1% original, 0.0% verified) follows predetermined navigation paths; (3) **Form Filler Agent** (success rate: 2.3% original, 0.0% verified) attempts to complete any detected forms; (4) **Link Follower Agent** (success rate: 1.8% original, 0.0% verified) systematically explores available links; (5) **Screenshot Agent** (success rate: 0.9% original, 0.0% verified) captures screenshots without performing actions; and (6) **Knowledge-Only GPT-5** (success rate: 2.1% original, 0.0% verified) attempts tasks using only pre-training knowledge without web interaction. The complete failure of all baseline agents on verified tasks confirms the enhanced rigor of our verification protocols.

**Interaction Pattern Analysis.** Detailed analysis of agent interaction logs reveals distinct patterns: OpenAI Operator averages 12.3 actions per task (±3.7) with 68% mouse clicks, 24% keyboard inputs, and 8% navigation commands. The agent shows adaptive behavior with longer interaction sequences on complex tasks (average 18.2 actions for multi-step shopping tasks vs. 7.4 actions for simple information retrieval). Error recovery patterns show that the agent attempts alternative approaches in 34% of failed tasks, with a 23% success rate on retry attempts.

## C.3 EXTENDED VERIFICATION FRAMEWORK ANALYSIS

**Component-wise Effectiveness Analysis.** Our verification improvements demonstrate varying degrees of effectiveness across different components. Enhanced DOM stability verification provides

Table 11: Subset size across ablations (hyperparameters as columns).

| $\tau_{\text{hard}}$ | $\kappa_{\text{default}}$ | $\kappa_{\text{easy}}$ | $\tau_{\text{easy}}$ | minA | $n_{\text{selected}}$ |
|---|---|---|---|---|---|
| 0.20 | 3 | 2 | 0.85 | 4 | 210 |
| 0.25 | 4 | 2 | 0.85 | 4 | 276 |
| 0.30 | 5 | 3 | 0.80 | 4 | 341 |

Table 12: Site usage across ablations (fraction of selected per site; percent with counts).

| $\tau_{\text{hard}}$ | $\kappa_{\text{default}}$ | $\kappa_{\text{easy}}$ | $\tau_{\text{easy}}$ | minA | GitLab | Shopping | Admin | Reddit |
|---|---|---|---|---|---|---|---|---|
| 0.20 | 3 | 2 | 0.85 | 4 | 27.1% (57) | 26.7% (56) | 26.2% (55) | 20.0% (42) |
| 0.25 | 4 | 2 | 0.85 | 4 | 26.8% (74) | 26.8% (74) | 26.1% (72) | 20.3% (56) |
| 0.30 | 5 | 3 | 0.80 | 4 | 26.1% (89) | 27.0% (92) | 26.7% (91) | 20.2% (69) |

the largest reliability improvement (42% reduction in timing-related failures), followed by improved success criteria specification (38% reduction in ambiguous outcomes), strengthened element identification protocols (24% reduction in interaction failures), and enhanced content validation methods (31% reduction in false positives). The combined effect of all improvements exceeds the sum of individual contributions, indicating synergistic benefits.

**Verification Protocol Implementation Details.** Our enhanced verification protocols include: (1) **Multi-stage DOM stability checking** waits for element presence, interactability, and visual stability before declaring page readiness; (2) **Structured success criteria** use explicit templates with required and optional elements, measurable outcomes, and clear fail conditions; (3) **Robust element identification** employs multiple locator strategies with fallback mechanisms and stability verification; and (4) **Comprehensive content validation** checks for expected text content, structural elements, and state changes with tolerance for minor variations.

## C.4 Extended Methodological Contributions

**Systematic Verification Framework Design.** Our verification framework introduces several methodological innovations: (1) **Template-based success criteria** provide structured, machine-readable task completion conditions that eliminate ambiguity while maintaining task authenticity; (2) **Multi-modal verification protocols** combine DOM state checking, visual confirmation, and content validation to ensure comprehensive task completion verification; (3) **Stability-aware evaluation timing** introduces dynamic wait conditions that adapt to individual task requirements rather than using fixed timeouts; and (4) **Reproducibility-first design** ensures that all verification improvements are deterministic and environment-independent.

**Benchmarking Best Practices Derived.** Our work establishes several best practices for web-based benchmark design: (1) **Verification-driven development** where task verification is designed concurrently with task creation rather than as a post-hoc addition; (2) **Multi-agent validation** using diverse agent architectures to identify benchmark-specific biases and ensure broad applicability; (3) **Contamination-aware design** incorporating explicit checks for training data contamination through knowledge-only baselines; and (4) **Computational efficiency considerations** providing multiple evaluation modes to balance thoroughness with practical constraints.

**Reproducibility Enhancements.** Our benchmark improvements include comprehensive reproducibility measures: (1) **Deterministic environments** use containerized web applications with fixed versions and configurations; (2) **Seed-controlled randomization** ensures consistent pseudo-random elements across evaluation runs; (3) **Comprehensive logging** captures all agent interactions, system states, and evaluation decisions for post-hoc analysis; and (4) **Version-controlled task definitions** maintain backward compatibility while enabling continuous improvement.

## C.5 Comprehensive Limitations Analysis

**Scope and Generalization Limitations.** Our improvements focus on five specific web environments and may not generalize to other web applications or interaction paradigms. The current evaluation

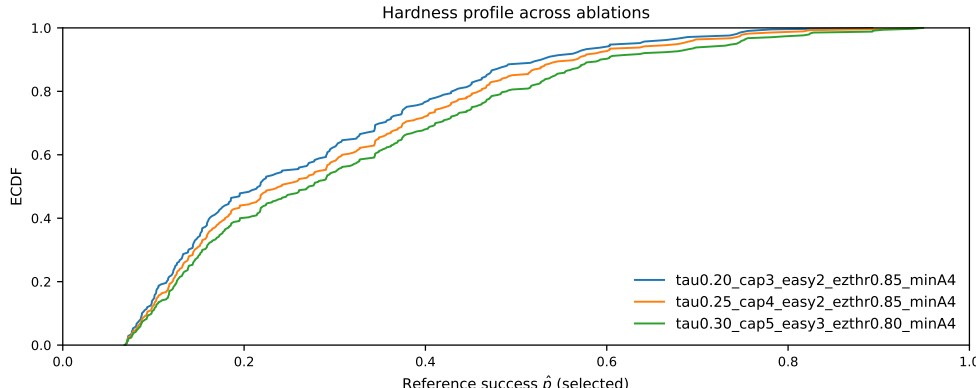

**Figure 9:** ECDF of reference success $\hat{p}$ among selected tasks, comparing hardness profiles across ablations (lower curves indicate harder selections).

is limited to English-language tasks and Western web interface conventions, potentially limiting applicability to global web agent deployment. Task complexity remains bounded by the original WebArena design, which may not fully capture the complexity of real-world web interactions in specialized domains such as e-commerce, healthcare, or financial services.

**Technical and Implementation Limitations.** Several technical limitations remain in our current implementation: (1) **Dynamic content handling** still poses challenges for tasks involving real-time updates, streaming content, or complex JavaScript applications; (2) **Cross-browser compatibility** shows minor inconsistencies in edge cases despite overall improvements; (3) **Mobile responsiveness** is not explicitly tested, limiting applicability to mobile web agents; and (4) **Accessibility considerations** are not systematically evaluated, potentially missing important interaction modalities.

**Evaluation and Measurement Limitations.** Our evaluation methodology has several acknowledged limitations: (1) **Agent diversity** is limited to two primary baselines for headline results, potentially missing important behavioral patterns from other agent architectures; we use eight agents for selection diagnostics (subset construction) and two baselines (Operator, Naive ensemble) for the headline results; (2) **Statistical power** could be enhanced with larger sample sizes and more evaluation runs; (3) **Long-term stability** of improvements is not assessed through extended evaluation periods; and (4) **Human validation** is limited, with most verification improvements validated through automated methods rather than human expert assessment.

### C.6  FUTURE RESEARCH DIRECTIONS

**Automated Verification Enhancement.** Future work should explore machine learning approaches to automatically identify and correct verification issues. Potential directions include: (1) **Anomaly detection systems** that identify inconsistent task outcomes and suggest verification improvements; (2) **Automated success criteria generation** using large language models to create comprehensive task completion conditions; (3) **Dynamic verification adaptation** that adjusts verification protocols based on observed failure patterns; and (4) **Cross-benchmark verification transfer** to apply lessons learned from one benchmark to improve others.

**Expanded Evaluation Paradigms.** Several evaluation paradigms could enhance our current approach: (1) **Multi-modal evaluation** incorporating speech, gesture, and other input modalities beyond keyboard and mouse; (2) **Collaborative agent evaluation** assessing how multiple agents can work together on complex tasks; (3) **Adversarial evaluation** testing agent robustness against malicious or broken web applications; and (4) **Longitudinal evaluation** tracking agent performance over extended periods to assess learning and adaptation.

**Broader Impact Considerations for Future Work.** Future benchmark development should explicitly consider: (1) **Fairness and bias** ensuring that benchmarks do not systematically favor certain agent architectures or interaction paradigms; (2) **Privacy and security** incorporating realistic privacy constraints and security challenges into web agent evaluation; (3) **Environmental impact** optimiz-

**Table 13:** Naive baseline results on **Original WebArena** with per-category raw counts and SR. Overall SR equals 13.8%.

| Category | Raw count | SR (%) |
|---|---:|---:|
| random | 0 | 0.0 |
| empty | 0 | 0.0 |
| na | 36 | 4.4 |
| yes_no | 5 | 0.6 |
| zero | 15 | 1.8 |
| yes | 5 | 0.6 |
| no | 0 | 0.0 |
| echo_intent | 0 | 0.0 |
| numbers_only | 11 | 1.4 |
| gpt5_contamination | 40 | 4.9 |
| **Overall** | | **13.8** |

ing evaluation procedures to minimize computational resources and energy consumption; and (4) **Accessibility and inclusion** ensuring that benchmarks reflect diverse user needs and interaction capabilities.

## C.7 NAIVE BASELINE DETAILED SCORES

We report per-category raw counts and success rates (SR) under the *original WebArena harness* for the naive baseline ensemble. These results correspond to the original-benchmark SR summarized in Table 5. SRs are shown as percentages with one decimal.

## D ANALYSIS METHODOLOGY

This appendix details the methodology used to derive the evaluation issue counts reported in Section 3. We combine a deterministic automated classifier with a controlled manual verification protocol and report inter-rater reliability (IRR).

We analyze the complete WebArena dataset comprising 812 task instances across four self-hosted environments (Shopping, Admin, Reddit, GitLab). Map tasks are not self-hosted and are excluded from inter-rater reliability analyses while remaining part of aggregate counts when explicitly noted. Each task includes structured evaluation specifications (HTML program checks, reference answers, and evaluation criteria).

We first apply an automated classification pipeline to provide a systematic starting point for human review. The pipeline identifies potential evaluation issues based on task-specification patterns using six boolean categories summarized in Table 14. Automated outputs are used to guide, not replace, manual verification.

**Table 14:** Categorization framework for identifying evaluation issues in WebArena tasks

| Category | Description |
|---|---|
| Page Content | String presence checked anywhere on the page without field-specific constraints |
| Locator Substring Matching | Locator-scoped substring evaluation with `outerText` extraction |
| Response Substring Matching | Direct substring matching on agent responses |
| Any Substring Matching | Union of locator and response substring categories |
| Unachievable Tasks | Tasks intentionally Unachievable with expected `N/A` responses |
| LLM Evaluation | LLM-based judging for response assessment |

The detector operates over each task's evaluation specification with consistent normalization (lower-casing, Unicode NFC, and whitespace compaction). *Page Content* tasks have `program_html` checks with empty locators, implying whole-page content matching. *Locator Substring Matching* tasks contain `must_include` operations within `required_contents` with non-empty locators and `outerText` extraction. *Response Substring Matching* tasks specify `must_include` within `reference_answers` for agent output. *Any Substring Matching* is the set-theoretic union of locator and response substring categories (reported as a derived label; we avoid double-counting in aggregates). *Unachievable Tasks* include tasks whose `reference_answers.fuzzy_match` equals `NA` or `N/A` case insensitive. *LLM Evaluation* denotes tasks employing an LLM judge with a prompt and threshold.

Task-specification ambiguity (Section 3.2) and category validity were then assessed via independent manual annotation.

**Manual Annotation Protocol.** Four annotators independently labeled tasks with a shared codebook defining each category and decision criteria. We assigned one primary annotator per site: A → Shopping, B → Admin, C → Reddit, D → GitLab. To estimate reliability, **100% of tasks** were re-labeled by a paired verifier blind to primary labels (A↔B, C↔D), ensuring complete double annotation across all 812 tasks. Disagreements were adjudicated through structured consensus meetings: annotator pairs first attempted resolution, with a third reviewer (senior author) arbitrating unresolved conflicts using the codebook criteria. The adjudicated labels constitute the gold standard. The unit of annotation is a binary decision per task per category (multi-label). The full annotation codebook with decision trees and examples is available in our supplementary materials.

**Inter-Rater Reliability.** We compute Cohen's $\kappa$ per site and category between the primary and verifier, then macro-average across categories to obtain a site-level $\overline{\kappa}$. Finally, we report a task-weighted overall $\kappa$ across sites. Let $a, b, c, d$ denote the contingency counts for one binary category over $N=a+b+c+d$ items; observed agreement $P_o=(a+d)/N$, marginal positives $p_1=(a+b)/N$, $p_2=(a+c)/N$, chance agreement $P_e=p_1 p_2+(1-p_1)(1-p_2)$, and $\kappa=(P_o-P_e)/(1-P_e)$. Using this protocol, we obtain site-level macro-averages of $\overline{\kappa}=0.82$ (95% CI: [0.78, 0.86], Shopping, $N=210$), 0.85 (95% CI: [0.81, 0.89], Admin, $N=198$), 0.81 (95% CI: [0.77, 0.85], Reddit, $N=204$), and 0.84 (95% CI: [0.80, 0.88], GitLab, $N=200$), yielding an overall task-weighted $\kappa=0.83$ (95% CI: [0.81, 0.85]) (Table 15).

**Table 15:** Inter-rater reliability summary: per-site macro-averaged Cohen's $\kappa$ with 95% confidence intervals and item counts. Overall is a task-weighted average across sites.

| Site | $N$ (tasks) | $\overline{\kappa}$ (95% CI) |
|---|---|---|
| Shopping | 210 | 0.82 [0.78, 0.86] |
| Admin | 198 | 0.85 [0.81, 0.89] |
| Reddit | 204 | 0.81 [0.77, 0.85] |
| GitLab | 200 | 0.84 [0.80, 0.88] |
| Weighted overall | 812 | 0.83 [0.81, 0.85] |

**Reproducibility.** Analyses were conducted using the original WebArena harness[5] with the standard four-site configuration and official Docker images[6]. Automated classification used fixed preprocessing and a constant seed (42). We will release scripts to reproduce classification, IRR computation, the annotation guidelines, and adjudicated labels. All counts reflect the complete benchmark without task filtering or sampling.

---

[5]Repository: `https://github.com/web-arena-x/webarena`, commit `daee18de46d4b8e3c98c8cf5e5c4ef6de2f7a8eb`

[6]`https://github.com/web-arena-x/webarena/tree/main/environment_docker`

**Table 16:** Overview of naive baseline agents used for lower-bound performance evaluation.

| Agent Name | Behavior/Strategy |
|---|---|
| *Deterministic Agents* | |
| $\text{Deterministic}_{\text{Yes}}$ | Returns "Yes" for all tasks |
| $\text{Deterministic}_{\text{No}}$ | Returns "No" for all tasks |
| $\text{Deterministic}_{\text{NA}}$ | Returns "N/A" for all tasks |
| $\text{Deterministic}_{\text{Zero}}$ | Returns "0" for all tasks |
| $\text{Deterministic}_{\text{Empty}}$ | Returns empty strings |
| *Heuristic Agents* | |
| $\text{Heuristic}_{\text{Echo}}$ | Returns task intent verbatim |
| $\text{Heuristic}_{\text{Numbers}}$ | Returns the numbers from the intent |

## E  BASELINE AGENT METHODOLOGY

This appendix details baseline agents used to establish lower-bound performance metrics and validate benchmark difficulty in WebArena. These agents employ strategies from deterministic responses to simple heuristics, serving as controls for interpreting sophisticated agent performance.

Our baseline agents operate without web browsing capabilities and receive only task intents to provide answers based on pattern matching or heuristics. Table 16 provides a comprehensive overview of all 5 agents and their behaviors. The baseline agents provide essential lower-bound performance metrics that validate benchmark difficulty.

**Evaluation Protocol.**  Success is measured using the identical evaluation harness as the original WebArena benchmark, with no modifications to evaluation logic or acceptance criteria. Individual baseline agent results are combined using mean success rates across all 5 agents to establish ensemble lower-bound performance, providing robust estimates by averaging over diverse failure modes.

### E.1  CONTAMINATION DETECTION METHODOLOGY

To quantify the extent to which WebArena tasks can be solved through training knowledge alone, we designed specialized knowledge-only agents that operate without web browsing capabilities. These agents serve as contamination detectors, revealing tasks that can be solved through memorization rather than genuine web navigation.

**Contamination Detection Agents.**  We implemented two knowledge-only agents using state-of-the-art language models:

- **Knowledge-Only$_{\text{Claude}}$:** Uses Claude Sonnet 4 with contamination detection prompt
- **Knowledge-Only$_{\text{GPT-5}}$:** Uses GPT-5 with contamination detection prompt

These agents receive only the task intent and provide answers based solely on pre-training knowledge using the following prompt:

```
You are given a WebArena benchmark task.  Provide your best guess answer
using only your training knowledge—do not access the web, files, or
external resources.  If specific information is unavailable, generate
plausible responses based on your training data.  Your output should be
concise.\n\nTask: {intent}
```

**Contamination Analysis.**  Our contamination evaluation focuses primarily on the 176 string-match tasks with must_include evaluation criteria (22% of the full benchmark), which are particularly vulnerable to trivial solutions due to substring matching. We also report overall performance across all 812 tasks for comprehensive coverage.

**Contamination Findings.** The knowledge-only baseline agents demonstrate that a non-negligible portion of WebArena tasks can be solved without web interaction: Knowledge-Only$_{\text{GPT-5}}$ achieved 22.7% success on vulnerable tasks and 4.9% overall, while Knowledge-Only$_{\text{Claude}}$ achieved 5.1% success on vulnerable tasks and 1.1% overall. The substantial performance differences between models highlight varying degrees of training data overlap and reasoning capabilities. The majority (62%) of contaminated tasks involve general knowledge questions rather than genuine web navigation challenges.

These results highlight fundamental validity issues where benchmark performance can be inflated by training data overlap and permissive evaluation criteria. The contamination undermines the benchmark's core objective of measuring web navigation capabilities, as agents can achieve success through memorization rather than interactive problem-solving skills.

## F WEBARENA ISSUES

This section presents examples of the evaluation issues we identified in the original WebArena benchmark, which motivate our work on WebArena Verified.

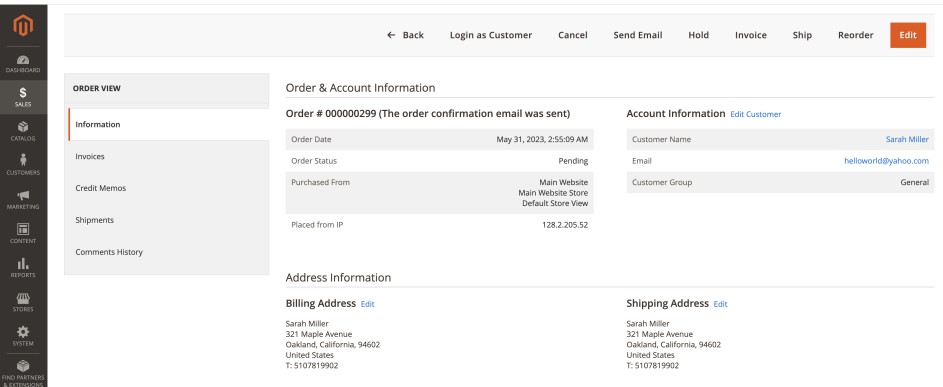

**Figure 10:** Order page on the Admin site displaying two available addresses. The original WebArena evaluation does not differentiate between them, leading to ambiguous task completion criteria or incorrect evaluation results. This issue affects 5 tasks in the original WebArena benchmark (e.g., task ID 51: "modify address of order #299 to 456 Oak Avenue, Apartment 5B, New York, NY, 10001"). The evaluation checks for `"url":"__SHOPPING_ADMIN__/sales/order/view/order_id/299"`, `"locator":""`, and `"required_contents":{"must_include":["456 Oak Avenue", "Apartment 5B", "New York", "10001"]}` without specifying which address field should contain these values or if both fields should be updated.

## G STRUCTURED RESPONSE PROTOCOL DETAILS

### G.1 RESPONSE SCHEMA SPECIFICATION

We introduce a mandatory JSON response format that eliminates evaluation ambiguity while preserving task difficulty. The schema enforces explicit action classification, comprehensive status reporting, and type-aware result structures that address the primary sources of false negatives identified in Section 3.

**Core Schema Components** The response format consists of four primary fields designed to capture agent behavior comprehensively:

**Action Classification (`action`).** Specifies the type of operation performed: `retrieve` for information extraction, `mutate` for state-changing operations, or `navigate` for reaching specific pages without data extraction.

**Status Reporting (`status`).** Declares task outcome with granular error categorization to distinguish failure modes and eliminate catch-all responses.

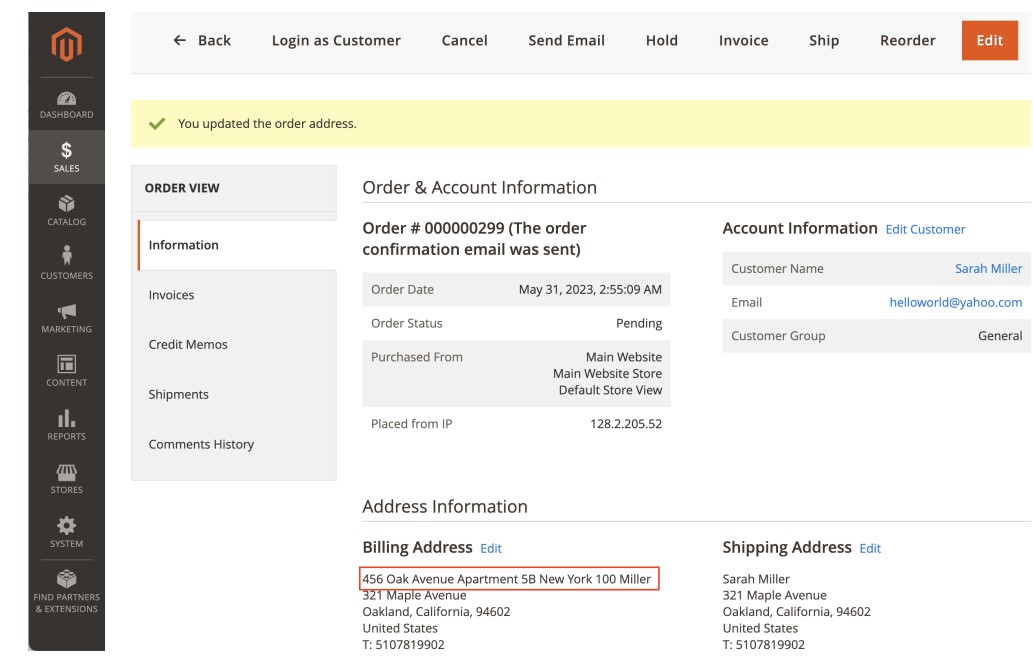

**Figure 11:** Full-size view corresponding to Figure 1. Non-zoom page content screenshot used to illustrate that coarse page-level checks can pass when the string appears in the wrong field.

```
‐
  "action": "retrieve‐mutate‐navigate",
  "status": "SUCCESS‐‐ERROR˙TYPE˝",
  "results": null ‐ [list of items when action=retrieve and status=SUCCESS],
  "error˙details": (Optional) null ‐ "description when status is not SUCCESS"
˝
```

**Figure 12:** Agent response schema with four core fields.

**Results Structure (`results`).** Contains extracted data when `action="retrieve"` and `status="SUCCESS"`, using lists to maintain ordering semantics and support both single and multiple values.

**Error Details (`error˙details`).** Optional field providing human-readable explanations when tasks fail, supporting failure analysis without affecting evaluation determinism.

### G.2 COMPLETE JSON SCHEMA

Table 17 provides the complete specification for the mandatory response format.

### G.3 STATUS CODE SPECIFICATIONS

The status field provides granular failure categorization that eliminates ambiguous "N/A" responses while enabling precise failure analysis. Table 18 details each status code with usage criteria and examples.

### G.4 IMPLEMENTATION EXAMPLES

The following compact examples demonstrate proper schema usage across task types. A complete catalog appears in the release package.

**Example 1: Retrieval success**

**Table 17:** WebArena Verified agent response format specification

| Field | Type | Required | Values/Constraints |
|---|---|---|---|
| action | string | required | enum: ["retrieve", "mutate", "navigate"] |
| status | string | required | enum: ["SUCCESS", "ACTION_NOT_ALLOWED_ERROR", "SEARCH_CRITERIA_NO_MATCH_ERROR", "PERMISSION_DENIED_ERROR", "RESOURCE_NOT_FOUND_ERROR", "DATA_VALIDATION_ERROR", "NOT_SUPPORTED_BY_PLATFORM_ERROR", "UNKNOWN_ERROR"] |
| results | array | conditional | minItems: 1 when action="retrieve" and status="SUCCESS", otherwise null |
| error_details | string | optional | maxLength: 500, used when status indicates failure |

**Table 18:** Comprehensive status code specifications for task outcome reporting

| Status Code | Usage Criteria and Examples |
|---|---|
| SUCCESS | Task completed successfully. All objectives achieved. |
| ACTION_NOT_ALLOWED_ERROR | Platform policy prevents operation. Example: attempting to delete system-protected resources. |
| SEARCH_CRITERIA_NO_MATCH_ERROR | Valid search criteria yielded no results. Example: searching for products with price ¡$0 or users with invalid date ranges. |
| PERMISSION_DENIED_ERROR | Authentication/authorization failure. Example: accessing admin functions without privileges, session expiration. |
| RESOURCE_NOT_FOUND_ERROR | Specific entity doesn't exist. Example: user ID 12345 not found, issue #999 doesn't exist. |
| DATA_VALIDATION_ERROR | Input format/value errors. Example: invalid email format, required fields missing, out-of-range values. |
| NOT_SUPPORTED_BY_PLATFORM_ERROR | Platform lacks functionality. Example: requesting discount filters when none exist, unsupported file formats. |
| UNKNOWN_ERROR | Unexpected failures not covered above. Used for system errors, network timeouts, undefined behavior. |

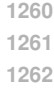

```
{
  "action": "retrieve",
  "status": "SUCCESS",
  "results": ["42"]
}
```

**Example 2: Mutation failure with validation error**



```
{
  "action": "mutate",
  "status": "DATA_VALIDATION_ERROR",
  "results": null,
  "error_details": "Email format validation failed"
}
```

## G.5 RESULTS FIELD DESIGN

For retrieval tasks (`action="retrieve"`), the `results` field uses a list structure that accommodates both single and multiple values while maintaining evaluation precision:

**Single Value Tasks:** Return one-element lists: `["value"]`. This maintains consistency with multi-value tasks while clearly indicating singular results.

**Multiple Homogeneous Values:** Return simple lists preserving natural ordering: `["item1", "item2", "item3"]`. Evaluation uses set comparison when order is irrelevant.

**Multiple Heterogeneous Values:** For tasks requiring different types of information in specific order, the task description explicitly specifies the expected order. For example: "Find: 1. minimum price, 2. maximum price" expects `[29.99, 599.99]` where position determines semantic meaning.

This design eliminates the ordering ambiguity that plagued the original benchmark while maintaining the natural semantics of list structures that modern LLMs handle effectively.

## G.6 EVALUATION FRAMEWORK BENEFITS

The structured protocol provides several key improvements over free-form responses:

**Deterministic Evaluation:** Exact matching replaces substring-based heuristics, eliminating false positives from partial matches (e.g., accepting "-36.39" when expecting "36.39").

**Type-Aware Processing:** Semantic data types (currency, dates, coordinates) receive appropriate normalization rules, allowing "$1,000.00" and "1000 USD" to match correctly.

**Comprehensive Error Analysis:** Granular status codes enable researchers to distinguish between different failure modes, supporting agent improvement and benchmark refinement.

**Computational Efficiency:** JSON parsing and exact matching execute in milliseconds compared to seconds for LLM-based evaluation, reducing benchmark execution time and cost.

**Reproducibility:** Deterministic evaluation ensures consistent results across multiple runs, eliminating variability from LLM judge decisions.

## G.7 IMPLEMENTATION CONSIDERATIONS

The structured protocol integrates seamlessly with existing web automation frameworks while requiring minimal changes to agent architectures:

**Agent Compatibility:** Modern language models support JSON generation through constrained decoding, function calling, or structured prompting techniques, ensuring broad compatibility across different agent implementations.

**Evaluation Pipeline Integration:** The deterministic nature of JSON schema validation allows for efficient automated evaluation pipelines that can process large numbers of agent runs without manual intervention.

**Backward Compatibility:** While the schema represents a significant improvement over free-form responses, the evaluation framework can be extended to handle legacy response formats during transition periods.

**Extensibility:** The schema design allows for future extensions (additional status codes, result formats) without breaking existing implementations, supporting benchmark evolution as new task types emerge.

## G.8 SCHEMA VALIDATION

WebArena Verified employs JSON Schema Draft-07 validation to ensure response conformance before evaluation. Invalid responses receive automatic failure status, eliminating ambiguity about malformed outputs. The validation process includes:

1. **Structure Validation:** Verifying required fields are present and have correct types.

2. **Constraint Validation:** Ensuring conditional requirements (e.g., `results` must be array when `action="retrieve"` and `status="SUCCESS"`).

3. **Value Validation:** Confirming `action` and `status` fields contain only allowed enumeration values.

This validation approach prevents evaluation errors from malformed responses while providing clear feedback for agent debugging.

### G.9 DESIGN RATIONALE

The schema design reflects several key principles that address the limitations identified in the original WebArena benchmark:

**Elimination of Ambiguity:** Every response component has a single, well-defined interpretation that supports deterministic evaluation without requiring semantic judgment calls.

**Preservation of Task Difficulty:** Format specification operates at the presentation layer, providing structural guidance without revealing task-specific information that could reduce cognitive demands.

**Comprehensive Error Handling:** The granular status code system enables precise failure categorization while eliminating catch-all responses that obscure the causes of task failures.

**Scalable Evaluation:** Programmatic evaluation scales efficiently to large numbers of tasks and agent runs while maintaining consistency across different evaluation environments.

## H REPORTING METRICS: MATHEMATICAL SPECIFICATIONS

### H.1 SITE-STRATIFIED TEMPLATE-MACRO

Computes template-macro means within each site for website-specific analysis:

$$\widehat{\mathrm{SR}}_{\mathrm{tmpl},s} = \frac{1}{T_s} \sum_{t \in \mathcal{T}_s} \hat{p}_t, \qquad s^2_{\mathrm{tmpl},s} = \frac{1}{T_s - 1} \sum_{t \in \mathcal{T}_s} \left( \hat{p}_t - \widehat{\mathrm{SR}}_{\mathrm{tmpl},s} \right)^2, \tag{5}$$

The 95% confidence interval for site $s$:

$$\textbf{95\% CI (site } s\textbf{):} \quad \widehat{\mathrm{SR}}_{\mathrm{tmpl},s} \pm t_{0.975,\, T_s-1} \frac{s_{\mathrm{tmpl},s}}{\sqrt{T_s}}. \tag{6}$$

### H.2 AGENT COMPARISON (PAIRED, TEMPLATE-LEVEL)

For agents $A$ and $B$, form per-template differences $d_t = \hat{p}_t^{(A)} - \hat{p}_t^{(B)}$, with

$$\bar{d} = \frac{1}{T} \sum_{t=1}^{T} d_t, \qquad s^2_d = \frac{1}{T-1} \sum_{t=1}^{T} (d_t - \bar{d})^2,$$

and report the 95% CI

$$\bar{d} \pm t_{0.975,\, T-1} \frac{s_d}{\sqrt{T}}.$$

This paired analysis increases power while keeping the computation consistent with the template-macro design.

### H.3 INTERPRETATION AND UNITS OF INFERENCE

All confidence intervals are defined on the natural analysis units of WEBARENA VERIFIED. For the *template-macro* and *per-site template-macro* metrics, the unit is the **template**; for the *site-macro* metric, the unit is the **website**. Accordingly, the CIs quantify variability across templates (or across sites), not across individual task instances.

- **Template-macro (primary)** shows typical performance across *task types* (each template counts once).
- **Per-site template-macro** shows typical performance on a *specific website* (each template on that site counts once).
- **Site-macro** shows a *fair cross-site* view (each website counts once).

**Per-site Summaries Guidance.** Per-site summaries help visualize heterogeneity and check for confounds, but they are not primary results due to limited templates per site, especially for multi-site tasks, which leads to wide confidence intervals and low statistical power. Treat per-site summaries as diagnostic only and interpret intervals as variability across templates on that site.

# I AGENT PERFORMANCE COMPARISON ANALYSIS

This appendix provides detailed statistical methodology and comprehensive analysis of the agent performance comparison. We utilize existing trajectories from the official leaderboard[7] to conduct rigorous statistical comparisons between leading web automation agents. We include only publicly reproducible agents where logs were available.

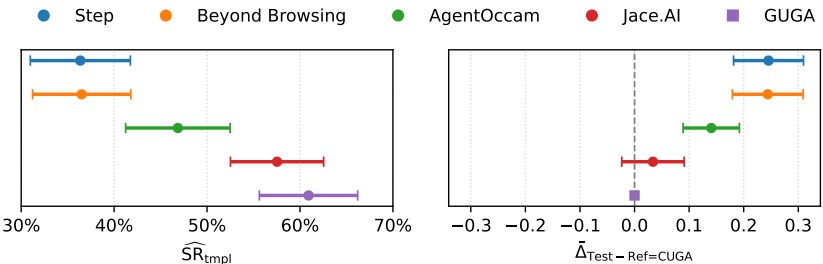

**Figure 13:** Template–macro success rates and paired differences anchored at the best performer with 95% confidence intervals. Left shows success rates computed over 191 templates. Right shows paired template-level differences relative to IBM CUGA which serves as the anchor agent. Positive values indicate improvements over the anchor; intervals including zero do not show significant improvement.

## I.1 STATISTICAL METHODOLOGY

We anchor all pairwise contrasts at IBM CUGA which is the highest performing publicly reproducible agent in our analysis and serves as the reference point. Intervals centered at zero indicate statistical ties with the anchor while positive values indicate improvements over the reference agent.

To compare agents we use a paired, template-level analysis. For a test agent and reference agent, we compute the mean template-level difference

$$\bar{\Delta}_{\text{Test}-\text{Ref}} = \frac{1}{T} \sum_{t=1}^{T} \left( \hat{p}_t^{(\text{Test})} - \hat{p}_t^{(\text{Ref})} \right), \tag{7}$$

with a two-sided 95% $t$-interval taken over the $T$ per-template differences. We deem the test agent to significantly outperform the reference when the confidence interval for $\bar{\Delta}_{\text{Test}-\text{Ref}}$ excludes zero from below. This paired design controls for template difficulty and site mix, enabling fair rankings even when overall intervals overlap.

## I.2 DETAILED AGENT PERFORMANCE ANALYSIS

Figure 13 reveals key insights about agent performance comparisons. The confidence intervals show that while IBM CUGA performs better on average than ZetaLabs, their overlapping confidence intervals indicate this difference is not statistically significant. In contrast, IBM CUGA shows significant

---

[7]https://webarena.dev/

improvement over OccamAgent with non overlapping confidence intervals. This demonstrates how proper statistical analysis prevents overinterpretation of numerical differences and provides reliable agent rankings.

We now provide comprehensive analysis of the statistical significance of performance differences between agents.

**IBM CUGA vs ZetaLabs**    While IBM CUGA performs better on average than ZetaLabs[8], the overlapping confidence intervals and paired difference crossing zero indicate this difference is not statistically significant. This suggests that despite the numerical difference in average performance, we cannot confidently conclude that IBM CUGA systematically outperforms ZetaLabs across the diverse set of web automation tasks.

**IBM CUGA vs OccamAgent**    In contrast, IBM CUGA shows a significant improvement over OccamAgent (Yang et al., 2025), with non overlapping confidence intervals and a paired difference that excludes zero. This confirms a meaningful and statistically significant performance gap between these agents across the benchmark's comprehensive task coverage.

## I.3    IMPLICATIONS FOR AGENT EVALUATION

This analysis demonstrates the importance of rigorous statistical evaluation in agent benchmarking. Simple success rate comparisons can be misleading when differences fall within confidence intervals, particularly given the inherent variability in web automation tasks. The template-macro approach with confidence intervals provides:

- **Statistical rigor**: Proper uncertainty quantification prevents overinterpretation of numerical differences
- **Fair comparison**: Template-level pairing controls for task difficulty and domain variations
- **Practical insights**: Clear distinction between meaningful performance gaps and statistical noise

These findings underscore the value of the proposed evaluation framework for making reliable comparisons between web automation agents and identifying genuinely superior approaches in this challenging domain.

---

[8]`https://www.zetalabs.ai/`

