# OpenReview forum: "WebArena Verified"
_ICLR.cc/2026/Conference — ICLR 2026 Conference Withdrawn Submission_

### Official Review · Reviewer_KeUD · 2025-10-19

**Soundness:** 3
**Presentation:** 4
**Contribution:** 2
**Rating:** 2
**Confidence:** 4

**Summary:**

This paper introduces WebArena Verified, a revised and enhanced version of the WebArena benchmark. The authors audited all 812 original tasks and their failure patterns, identifying flaws such as fragile evaluation logic, ambiguous instructions, and knowledge contamination. They propose a new version, WebArena Verified, that addresses these flaws. They also introduce WebArena Verified Hard, a smaller and more challenging subset of the verified dataset. Using the OpenAI Operator, they show that the new dataset strengthens evaluation while preserving WebArena's realism.

**Strengths:**

1. The authors conducted a rigorous study of both the original WebArena benchmark and their revised version, as well as a detailed analysis of agent failure patterns. The lessons learned are valuable for the designers of future benchmarks.

2. The authors proposed several new verification methods that successfully reduce many of the false positives and false negatives present in the original benchmark's design.

3. The authors provide a newly revised version of the WebArena dataset and a carefully selected "hard" subset, which will be valuable to the community.

**Weaknesses:**

1. While "Type-Aware Exact Matching" is an improvement over naive string comparison, I am not convinced it can fully replace LLM-based judging. It is difficult to normalize dates, currencies, and coordinates in a way that covers all possible edge cases.

2. The "Playwright network monitoring" feature aims to detect knowledge contamination, where agents use pre-existing knowledge to complete a task without interacting with the website. However, this may unfairly penalize two legitimate agent designs: a) agents that persist memory across tasks, and b) agents that use API calls to interact with websites.

3. The new requirement for a "structured action field" gives the harness a greater capability to assess task understanding, but it also introduces the additional challenge of formatting structured responses. The ability to formulate these responses goes beyond the original testing scope of WebArena.

4. Finally, the paper's contribution seems somewhat limited. While the systematic analysis of WebArena's failure patterns is valuable, this contribution is partially attributed to the *Establishing Best Practices for Building Rigorous Agentic Benchmarks* paper that this work cites. The primary contribution is the new dataset, which is an incremental improvement on an existing benchmark.

**Questions:**

1. Why not open-source the code, data, and tools during the review phase via an anonymous repository? This would help reviewers better understand the contribution.

2. Did you reach out to the original WebArena team? If so, have they acknowledged the flaws that you systematically discovered?

---

### Official Review · Reviewer_FSHA · 2025-10-30

**Soundness:** 3
**Presentation:** 3
**Contribution:** 1
**Rating:** 4
**Confidence:** 4

**Summary:**

The authors develop WebArena Verified, a modification of WebArena that addresses issues in the current benchmark, including repairing checks and clarifying instructions. They demonstrate that this leads to a significant reduction in the false negative rate when evaluating agents. They additionally release a smaller hard subset of WebArena Verified, reducing evaluation time but retaining discriminatory power.

**Strengths:**

- Improves robustness of agent evaluation
- Removal of many LLM-as-judge requirements improves determinism

**Weaknesses:**

- Benchmarks and modifications thereof are useful when they tell us something new about models and their capabilities. It would be interesting to compare the accuracy achieved on WebArena versus WebArena Verified for more models, and to determine if there is a substantive change in the relative rankings of the models. Currently, evaluation is lacking, which limits our understanding of the benefits of WebArena Verified vs WebArena.
- Lack of novelty: this paper is at most an incremental change over existing WebArena, and I am unsure whether data-cleaning of an existing benchmark merits its own paper.
- Claims of discriminatory power of the hard subset are not supported in the main text, as there are far too few models compared.

**Questions:**

- Could you perform evaluations to see the scores and relative rankings of more models on WebArena vs WebArena Verified?
- On what grounds are you claiming that the hard subset retains discriminative power?

---

### Official Review · Reviewer_QCGH · 2025-10-30

**Soundness:** 2
**Presentation:** 2
**Contribution:** 2
**Rating:** 2
**Confidence:** 3

**Summary:**

The paper revisits the WebArena benchmark and proposes WebArena Verified, a re‑evaluation that preserves the original containerized environments while substantially redesigning the measurement pipeline. Through a full audit of 812 tasks, the authors fix several problems (e.g., misalignment and ambiguities, brittle string matching) . Authors also release a difficulty‑focused 258‑task subset (WebArena Verified Hard) claimed to retain discriminative power while reducing runtime.

**Strengths:**

The paper makes a useful contribution by addressing reliability issues in WebArena. The authors conduct a full audit of 812 tasks and introduce type-aware matching, backend state checks, and structured JSON outputs. These changes reduce false positives and negatives and improve determinism in evaluation. The work is timely and aligns with best practices from other verified benchmarks like SWE-bench Verified and OSWorld-Verified. Overall, the design improvements are well motivated and clearly documented.

In summary,
- The paper directly tackles important reliability issues in widely used WebArena benchmark.
- All 812 tasks are examined under a documented protocol with double annotation and good inter‑rater agreement, yielding 81 reference alignment fixes and 218 ambiguity resolutions. This shows depth and rigor rather than anecdotal corrections

**Weaknesses:**

**Summary:**
- The work is incremental and the impact is narrow.
- The evaluation relies mainly on a single agent and one seed. This limits generalizability and makes claims about ranking stability and performance improvements weak. Multi-agent, multi-seed experiments are essential for a benchmark revision.
-  Improvements target string‑verifiable or state‑verifiable tasks within five sites; limits include English‑only tasks, desktop web conventions, and missing state traces in historical logs—constraining generalization to other environments and modalities
- The proposed “Verified Hard” subset fails pre-registered stability thresholds (Kendall τ and MAE). This undermines its intended role as a reliable low-cost proxy. Without stability, the subset risks misleading comparative studies.
- While network‑activity checks remove trivial no‑browsing successes, the paper admits agents can still navigate to a page and answer from model memory without grounding in retrieved content; no page‑content attestation or citation‑based grounding is implemented

**Details:**
The paper makes a useful contribution by addressing reliability issues in WebArena. The authors conduct a full audit of 812 tasks and introduce type-aware matching, backend state checks, and structured JSON outputs. These changes reduce false positives and negatives and improve determinism in evaluation. The work is timely and aligns with best practices from other verified benchmarks like SWE-bench Verified and OSWorld-Verified. Overall, the design improvements are well motivated and clearly documented.

However, the contribution of the paper is narrow. It proposes a better version of WebArena, which is only one of the many benchmarks out there now. The work does not benefit users of other benchmarks. It would be more interesting if authors could show how their methodology could be generalized, with (semi-) automated techniques, to other benchmarks as well.

Another major concern is the narrow scope of empirical validation. Most results rely on a single agent (OpenAI Operator) and one seed, as noted in Section 5 and Appendix A. This limits confidence in claims about ranking stability and generalizability. Related work such as SWE-bench Verified emphasizes multi-agent, multi-seed evaluations to ensure robustness. I recommend expanding experiments to include diverse agent families and multiple seeds to strengthen the evidence base.

The proposed “Verified Hard” subset does not meet its stated stability goals. Table 8 shows Kendall τ and MAE thresholds are not satisfied, and Table 10 reports high selection variability (mean selection probability ≈0.30). This undermines the claim that the subset preserves rankings and could mislead comparative studies. To improve, consider increasing subset size, applying stratified sampling, or reframing the subset as a diagnostic stress test rather than a ranking proxy.

The paper acknowledges that agents can still answer from memory without using retrieved content (Section 5.2). While network-activity checks prevent trivial no-browsing successes, they do not guarantee evidence-based answers. This gap is critical because contamination and hallucination remain major concerns in web-agent evaluation. We suggest adding DOM span or citation-based attestation, similar to pointer-based evaluation in open-domain QA, to enforce grounding and improve reliability.

Some methodological choices lack sufficient justification. For example, the survival-style GLMM for hardness and policy caps (κ_default, τ_hard/easy) appear hyperparameter-driven, and sensitivity analyses in Appendix B show variability in selection outcomes. This raises questions about reproducibility and robustness. We recommend providing clearer rationale for these choices and adding ablation studies or alternative models to demonstrate that conclusions are not overly dependent on specific parameter settings.

**Questions:**

1. Could you explain why the “Verified Hard” subset fails Kendall τ and MAE thresholds? Do you plan to adjust the subset size or selection method to improve stability?
2. Why were most evaluations conducted with a single agent and one seed? Do you have results for other agent families or multiple seeds to confirm robustness?
3. How does the benchmark ensure that answers are based on retrieved content rather than model memory? Are there plans to add DOM span or citation-based checks?

---

### Official Review · Reviewer_8zd3 · 2025-10-31

**Soundness:** 2
**Presentation:** 2
**Contribution:** 2
**Rating:** 2
**Confidence:** 3

**Summary:**

- This paper introduces WebArena Verified, a re-evaluation of WebArena that reduces the false-negative rate by 11.3 percentage points compared to Web Arena.
- The authors also introduce WebArena Verified Hard, a subset of the full tasks that retains difficult tasks and reduces runtime.

**Strengths:**

- The paper conducts a systematic audit of WebArena tasks in terms of task specification and evaluation reliability.
- Significant engineering effort seems to have been done to turn WebArena into WebArena Verified, by introducing several concrete technical improvements to address issues in Web Arena

**Weaknesses:**

- The work focuses exclusively on the WebArena benchmark. As such, it is unclear how the methodology could be systematically applied to other benchmarks. A clearer framework for generalization would strengthen its broader impact.
- The experimental validation only has 2 agents: OpenAI Operator and a simple baseline ensemble. Evaluating multiple diverse agents or LLMs would better demonstrate WebArena Hard and WebArena Verified Hard discriminative power and stability across different architectures.
- Constructing a “hard” version is not novel (e.g., https://arxiv.org/abs/2406.11939). The authors could more clearly state how their approach is similar or different from others.

**Questions:**

Please address each of the weaknesses.

---

### Note · Authors · 2025-11-19

I have read and agree with the venue's withdrawal policy on behalf of myself and my co-authors.